# Persistent Nasal Obstruction: An Expression of the Pro-Inflammatory State?

**Fernando M. Calatayud-Sáez [1],[*], Blanca Calatayud [2] and Ana Calatayud [3]**

1 Pediatrician at the Child and Adolescent Clinic "La Palma" Ciudad Real, 13001 Castilla la Mancha, Spain
2 Nutritionist at the Child and Adolescent Clinic "La Palma" Ciudad Real, 13001 Castilla la Mancha, Spain; blanca.calatayud@gmail.com
3 Nurse and Nutritionist at the Child and Adolescent Clinic "La Palma" Ciudad Real, 13001 Castilla la Mancha, Spain; anacalatayud94@gmail.com
* Correspondence: altayud@gmail.com

**Abstract:** Introduction: During the first years of life, the oro-pharyngeal lymphoid tissue gradually increases in size, causing in some children difficulty breathing and often leading to surgical removal of the tonsils and adenoids. The objective of the study is to assess the effects of the Mediterranean diet in children who had chronic upper airway obstruction. Material and methods: This was a prospective study pre-test/post-test comparison. Eighty-seven patients from two to eight years old were recruited. A food reeducation program based on the Mediterranean diet was applied for one year. Clinical, therapeutic, and anthropometric variables were studied. Results: The degree of nasal obstruction decreased in 95.1% of the patients. After the nutritional intervention, the number of colds with bacterial complications decreased by 80.26%; 60.9% had no bacterial complications during the year of the study. The use of antibiotics decreased by 81.94%. Symptomatic treatment decreased by 61.2%. Most patients did not require surgical intervention, and clinical evolution suggested that it would no longer be necessary. Conclusions: We can conclude by saying that the application of the traditional Mediterranean diet could be effective in the prevention and treatment of persistent nasal obstruction, limiting pharmacological and surgical intervention in many of these patients.

**Keywords:** persistent nasal obstruction; chronic obstruction of the upper airway; tonsillar hypertrophy; adenoid hypertrophy; pro-inflammatory state; Mediterranean diet; dietary intervention



## 1. Introduction

The tonsils and adenoids are complex structures related to the mucosa-associated lymphoid tissue (MALT), and are part of Waldeyer's ring, just at the entrance of the air and digestive tracts. Due to their anatomical location, they represent the first contact zone of microorganisms and antigenic substances existing in the inhaled air and in food. After approximately six years, they reach the maximum volume, and later they decrease progressively. When they reach adulthood, they almost disappear [1]. The hyperplasia of the tonsils and adenoids is closely related to various concomitant pathologies of neighboring structures, such as upper respiratory tract infections (URI), tonsillitis, adenoiditis, acute otitis media, otitis media with chronic exudate, rhino-sinusitis, and obstructive sleep apnea [2]. In primary care pediatrics, we see how many patients who come to the consultation from birth experience enlarging of these lymphoid structures, almost obstructing the oropharynx and rhinopharynx and developing a progressive obstructive pathology that prevents or hinders correct feeding and free air breathing, as well as anomalies in the conciliation of sleep [3]. All of this affects the quality of psychosocial life, school absenteeism on the part of the child and work of the parents, and increased spending on medicines, emergency assistance, hospitalization, and surgical intervention [4,5]. Several pharmacological treatments have been recommended, such as the application of antihistamines and decongestants, preventive antibiotics over a prolonged period [6], bacterial autovaccines [7], oral and

nasal corticosteroids, anti-inflammatory drugs, such as Montelukast, with controversy and generally insignificant results [8,9]. The proposed surgical treatment sponsored by ENT specialists can vary between adenoidectomy, placement of tympanic drainages, and tonsillectomy, depending on the case, especially when it is associated with tubal obstruction and obstructive sleep apnea [10,11], and is somewhat of a failure of pediatric preventive medicine, which has not been able to prevent the excessive growth of tonsils and vegetations [12–14]. There are more studies that demonstrate the anti-inflammatory effects of the Mediterranean diet [15], which has allowed us to develop the hypothesis that recurrent inflammatory episodes in childhood are closely related to the abandonment of a traditional diet. In this line of research, in which we have been working for years, we observed that the Mediterranean diet positively influences the control of the inflammatory and immune system, significantly improving the response to upper respiratory tract infections and significantly reducing their complications [16,17]. The objective of the study is to assess the effects of the traditional Mediterranean diet (TMD) during one year of application in children who presented persistent nasal obstruction.

## 2. Material and Methods

1. Study design: An analytical, prospective comparison study, pre-test/post test, was carried out. The sample included, consecutively, all patients from two to eight years old who presented with symptoms of persistent nasal obstruction (PNO), also called chronic upper airway obstruction, during the period from September 2009 to October 2018. The parents or guardians of the patients with PNO were asked to participate in the study, involving the whole family in the nutritional treatment, for which informed consent was requested. The study consisted of comparing the previous state and the subsequent situation after one year after having applied the TMD, being monitored on the nutritional education program "Learning to eat from the Mediterranean", which we have already used in previous studies [16,17]. The study was approved by the research committee of the General University Hospital of Ciudad Real (Internal code: C-95, Act 03/2017).

2. Measurement of the parameters of clinical and therapeutic evolution: The main variable was the presence of persistent nasal obstruction. The following secondary variables were considered: the number of upper respiratory tract infections (URI), the number of bacterial complications, otitis media with effusion, assistance in the emergency department, the number of symptomatic treatments, and the number of prescribed antibiotics, all valued by person and year. A clinical assessment was made on the degree of involvement of PNO: (mild = 0, moderate = 1, or intense = 2). Finally, a basic otorhinolaryngological exploration was performed, including rhinoscopy, pharyngoscopy, otoscopy, tympanometry with Microtymp'3 by Welch Allyn®, audiometry of the children collaborators (Audioscope by Welch Allyn®), intentional assessment of the face (adenoid facies), and cervical adenopathies [8]. PNO was defined as the persistent difficulty to nasally breathe adequately with associated respiratory symptoms, such as mouth breathing, snoring, difficult breathing in sleep, respiratory arrest during sleep (apneas), restless sleep, hyperflexion postures of neck to be able to sleep, drowsiness or feeling of not having rested properly, adenoid facies, and difficulties in swallowing [18–20]. An episode of URI was defined by two or more of the following criteria: fever greater than 38 °C measured with a tympanic thermometer, nasal congestion or mouth breathing, nasal discharge, odynophagia, and cough. The criteria for acute otitis media were based on the American Pediatric Association's Guide: (1) acute presentation; (2) presence of exudate in the middle cavity of the ear demonstrated by tympanic bulging, tympanometry, or otorrhea; (3) inflammatory signs and symptoms, such as otalgia or evident reddening of the eardrum [21]. Rinosinusitis was defined following the protocols of the Spanish Association of Pediatrics to the persistence of diurnal cough or rhinorrhea for more than ten days, without apparent improvement, in the context of an upper respiratory infection [22]. Otitis media with chronic exudate was considered when the exudate or bilateral effusion persisted for more than three months or more than six if it was unilateral. In patients suspected of allergic processes, tests were

performed to rule these out. In the last years of the study, and therefore to a small number of patients (15), analytical measurements were made with inflammatory markers (hsCR, IL-1, IL-6, TNF-alpha, C-3, and C-4).

3. Index of clinical and therapeutic evaluation performed to parents or guardians. To assess the clinical evolution of patients, we designed a questionnaire aimed at parents or guardians, in which symptoms related to persistent nasal obstruction, such as nasal breathing, sleeping noises, falling asleep, and night rest, were evaluated, as well as the recurrent colds and their complications, the intensity of the clinical pictures, the tolerance and difficulties with the diet performed, and the degree of satisfaction with the intervention. Each question in the questionnaire could be answered regarding the improvement observed with: 3 = a lot, 2 = quite, 1 = something, 0 = nothing (Table 1). Ten questions about the clinic and treatment were evaluated in the last four weeks, and they were scored between 0 (poor control) and 30 (good control). A patient was considered to be poorly controlled when he had a score equal to or less than 10.

**Table 1.** Clinical and therapeutic evaluation index. Responses from parents or guardians regarding the improvement observed: 3 = a lot, 2 = quite, 1 = something, 0 = nothing.

|  | Four Months | One Year |
|---|---|---|
| Do you breathe better through your nose? | 2.21 | 2.50 |
| Is there less noise when breathing? | 2.47 | 2.90 |
| Do you have sleep apnea or respiratory stops at night? | 2.51 | 2.90 |
| Do you rest better at night? | 2.65 | 2.90 |
| Has the number of colds decreased? | 2.40 | 2.90 |
| Have you noticed less intensity in the infectious processes? | 2.35 | 2.90 |
| Have complications decreased? | 2.65 | 2.90 |
| Has there been good diet tolerance on the part of the patient? | 2.70 | 2.90 |
| Has food quality improved? | 2.66 | 2.90 |
| Are you satisfied with the results? | 2.60 | 2.90 |

4. Parameters of weight statural evolution: By limiting foods that are part of the new western food culture, we have evaluated the correct weight statural development of the patients included in the study. To do this, we collected anthropometric data, such as weight, height, skinfolds, arm perimeters, abdomen, and waist circumferences, and with them, we calculated the body mass index, lean mass, and body fat mass [23].

5. Parameters of adherence to the TMD: To evaluate the dietary habits of patients and their families, we used the KidMed test [24,25] and the TMD test that we presented in previous works with the intention of covering the proposed changes by the TMD [16,17].

6. Foundations of the traditional Mediterranean diet: This diet is characterized by a high content of fresh, raw, perishable, and seasonal foods rich in vegetable fiber, minerals, vitamins, enzymes, and antioxidants; abundancy of fruits, vegetables, legumes, and whole grains, the characteristics of which include a low to moderate glycemic index; sufficient polyunsaturated fats from crude oils, nuts, seeds, and fish; low protein and saturated fat content of animal origin; and a low use of precooked and industrial foods. This means, in daily practice, the limitation of products such as white bread, industrial pastries, cow's milk, red and processed meats, sugary industrial beverages, and precooked fast food [26]. The TMD is based on the Decalogue that the Foundation of the Mediterranean Diet proposes on its website (Table 2) [27].

**Table 2.** The Mediterranean diet.

| 10 Basic Recommendations |
| --- |
| 1. Use olive oil as your main source of added fat. |
| 2. Eat plenty of fruits, vegetables, legumes, and nuts. |
| 3. Bread and other grain products (pasta, rice, and whole grains) should be a part of your everyday diet. |
| 4. Foods that have undergone minimal processing and are fresh and locally produced are best. |
| 5. Consume dairy products on a daily basis, mainly yogurt and cheese. |
| 6. Red meat should be consumed in moderation and, if possible, as part of stews and other recipes. |
| 7. Consume fish abundantly and eggs in moderation. |
| 8. Fresh fruit should be your everyday dessert, and sweets, cakes, and dairy desserts should be consumed only on occasion. |
| 9. Water is the beverage par excellence in the Mediterranean diet. |
| 10. Be physically active every day, since it is just as important as eating well. |

This has been proclaimed a cultural heritage and intangible heritage of humanity by Unesco [28]. In Table 3, we expose the differences between the TMD and the diet promoted by Western civilization.

**Table 3.** Differences between the traditional Mediterranean diet and the Western civilization diet.

| Traditional Mediterranean Diet | Western Civilization Diet |
| --- | --- |
| • Breastfeeding | • Adapted milk |
| • Varied, seasonal fruit | • Baby food jars and canned fruits |
| • Vegetables and leafy vegetables | • Baby food jars and canned and leafy vegetables |
| • Pulses and non-processed nuts | • Canned pulses and dried, fried, or salted nuts |
| • Minimally processed and fermented whole grains | • Refined, processed cereals with industrial fermenting agents |
| • Fermented milk, principally goat and sheep | • Whole, processed milks, mainly from cows |
| • Occasional lean meat, in small quantities | • High consumption of red, processed meats |
| • Minimally processed, perishable, fresh, and local foods | • Nonperishable processed and ultra-processed foods |
| • Limits on products with added chemicals | • Presence of chemical agents and enzyme disrupters |

7. Sample size and statistical analysis: To calculate the sample size, a significance level of 0.05 and a power of 80% were used, assuming a decrease in the degree of involvement of PNO per patient and per year of one unit and a standard deviation of 3.5 units, adjusting to a 25% loss, which resulted in a sample size of 80 patients. For the analysis of the results, the statistical package SPSS 15.0 was used. A descriptive analysis was carried out with statistics of central tendency and dispersion for quantitative variables and absolute and relative frequencies for qualitative variables. The comparison of the results of the different variables before and after the intervention was carried out by means of the Student's *t*-test for paired data when the variables followed a normal distribution or by the Wilcoxon test when they did not adjust to normal, after checking with the Shapiro–Wilk test.

## 3. Results

Through consecutive sampling, 98 patients were proposed to participate in the study as we were diagnosing PNO in the indicated period, of whom three refused to participate and eight left the program before the third session (two due to social difficulties in implementing the diet, another for disagreement with the elimination of some foods, and five on the recommendation of ENT specialists not coordinated with our team who operated on them surgically). The study was completed by a total of 87 patients, of whom 40 were girls and 47 boys, with an average age of 4.6 years. All of the patients included in the study were evaluated at the beginning and at four and 12 months after the start of nutritional therapy. Table 4 shows the characteristics of the sample. The results obtained were similar in both sexes, which is why they are collected together.

**Table 4.** Sample characteristics. Average age = 4.6 years.

|  | Boys (*n* = 47) | Girls (*n* = 40) |
|---|---|---|
| Weight (kg) | $20.39 \pm 3.17$ | $20.53 \pm 2.25$ |
| Height (m) | $0.96 \pm 0.11$ | $1.07 \pm 0.06$ |
| BMI (kg/m$^2$) | $16.44 \pm 1.28$ | $16.97 \pm 1.44$ |
| Fat mass (%) | $15.23 \pm 2.95$ | $17,59 \pm 2.68$ |
| Lean mass (%) | $84.77 \pm 4.32$ | $82.41 \pm 1.93$ |

Table 5 shows the results of the degree of involvement of PNO, the presence of recurrent diseases and their complications, and the pharmacological treatment of patients in the year prior to diagnosis and after the introduction of the TMD. The main variable (PNO) decreased considerably from 1.92 (moderate to intense) to 0.26 (non-mild). Likewise, the secondary variables improved, generally reducing clinical symptoms and pharmacological treatment.

**Table 5.** Evolution during the previous year and during the year of treatment.

|  | Previous Year | Year | *p* |
|---|---|---|---|
| Degree of involvement of persistent nasal obstruction (PNO); 0 = mild, 1 = moderate, 2 = intense | $1.92 \pm 0.27$ | $0.26 \pm 0.05$ | 0.001 |
| Number of URIs (upper respiratory infections) | $4.83 \pm 1.41$ | $1.93 \pm 0.37$ | 0.016 |
| Other complications | $3.09 \pm 0.75$ | $0.61 \pm 0.06$ | 0.01 |
| Children with otitis media with effusion (OME) associated with PNO | 47% | 1% | 0.001 |
| Number of emergency treatments | $1.52 \pm 0.76$ | $0.2 \pm 0.04$ | 0.01 |
| Antibiotics | $2.88 \pm 0.37$ | $0.52 \pm 0.08$ | 0.012 |
| Symptomatic treatment | $5.59 \pm 0.96$ | $2.17 \pm 0.39$ | 0.021 |

The anthropometric variables at the beginning, at four months, and after the intervention are shown in Table 6. In the determinants of growth and development, such as height and lean mass, an adequate, statistically significant increase was produced. The increase in average weight the year before the study was 2.11 kg compared to the current 2.93 kg, and the increase in average height was 6.5 cm compared to the current 6.9 cm. The body mass index (BMI) decreased slightly. The area of lean mass of the arm increased slightly, while the area of fat mass decreased discreetly.

**Table 6.** Anthropometric assessment at the start, after four months, and after one year.

|  | Start | Four Months | One Year | *p* |
|---|---|---|---|---|
| Weight (kg) | 20.45 ± 2.10 | 21.25 ± 1.89 | 23.38 ± 3.97 | 0.001 |
| Height (m) | 1.07 ± 0.35 | 1.10 ± 0.21 | 1.14 ± 0.32 | 0.001 |
| BMI (body mass index) | 16.88 ± 1.71 | 16.46 ± 1.12 | 16.45 ± 2.33 | 0.19 |
| Fat mass (%) | 16.71 ± 2.94 | 16.04 ± 3.83 | 17.01 ± 3.20 | 0.21 |
| Lean mass (%) | 83.29 ± 1.18 | 83.96 ± 1.98 | 82.99 ± 1.99 | 0.019 |

The average value of the KidMed index evolved from a score considered medium-high at the beginning of the program to an optimal value at the end (Table 7).

**Table 7.** KidMed test (percentage).

|  | Start | Four Months | One Year |
|---|---|---|---|
| One piece of fruit per day | 88.5 | 97.7 | 87.8 |
| One+ piece of fruit per day | 24.1 | 66.7 | 74.5 |
| One vegetable per day | 78.2 | 92.0 | 84.7 |
| Vegetables more than once per day | 4.6 | 46.0 | 60.2 |
| Regularly eats fresh fish (2–3 times/week) | 80.5 | 88.5 | 82.7 |
| Visits fast food restaurant once or more per week | 16.1 | 2.3 | 1.0 |
| Legumes 1–2 times/week | 90.0 | 90.0 | 88.8 |
| Pasta and rice every week | 95.4 | 98.9 | 87.8 |
| Cereal or derivative for breakfast | 95.4 | 97.7 | 86.7 |
| Regularly eats dried fruit and nuts | 6.9 | 35.6 | 52.0 |
| Olive oil used at home | 83.9 | 97.7 | 86.7 |
| No breakfast | 14.9 | 1.1 | 1.0 |
| Dairy at breakfast | 83.9 | 97.7 | 86.7 |
| Factory-baked goods for breakfast | 42.5 | 6.9 | 1.0 |
| Two yoghurts or 40 g of cheese/day | 96.6 | 100.0 | 88.8 |
| Sweets and snacks every day | 48.3 | 3.4 | 2.0 |

The TMD test evolved from levels considered low quality to optimal levels (Table 8 and Figure 1).

**Table 8.** Traditional Mediterranean diet test (%).

|  | Start | Four Months | One Year |
|---|---|---|---|
| Minimum of two pieces of fruit every day. | 49.4 | 78.2 | 94.3 |
| Fresh vegetables at every meal, as a first course, or as part of the main course. | 40.2 | 51.7 | 78.2 |
| Limited sugar intake (sweetened breakfast cereal, sweetened yoghurts or milkshakes, cakes, soft drinks, sugary biscuits, sweets, ice cream, etc.). | 10.3 | 73.6 | 75.9 |
| Sporadic use of potatoes (1–2 times/week) and preferably not fried. | 25.3 | 77.0 | 85.1 |
| Enjoys legumes and eats them one or more times a week, not always accompanied by meat. | 37.9 | 81.6 | 80.5 |
| Regular intake of white fish, oily fish, and seafood (2–3 times/week). | 69.0 | 72.4 | 92.0 |
| Consumes whole grains (whole wheat pasta, brown rice, whole wheat bread, etc.) in a controlled way and limits the consumption of refined flour, such as white bread, to less than 40 g per day). | 19.5 | 59.8 | 82.8 |
| Limits the consumption of preservatives and hydrogenated vegetable fats, regularly using unprocessed homemade foods. | 26.4 | 71.3 | 83.9 |
| Dairy: ingested, preferably skimmed in the form of natural yogurt, and preferably goat or sheep cheese, avoiding the use of sugary yogurts, dairy desserts, creams, margarines, ice creams, etc. | 12,6 | 65,5 | 85,1 |
| Only lean processed meats, less than twice per week. | 14.9 | 64.4 | 82.8 |
| Preferably white meat, less than three times per week (lean). | 26.4 | 71.3 | 82.8 |
| 30–50% of daily menu consists of raw or undercooked foods (fruits, vegetables, greens, soups, purees, raw nuts, extra virgin olive oil, etc.), preferably choosing seasonal ones. | 4.6 | 28.7 | 62.1 |

**Table 8.** *Cont.*

| | Start | Four Months | One Year |
|---|---|---|---|
| Junk food (indoors or outdoors) no more than one time per week. | 42.5 | 67.8 | 69.0 |
| Consumes extra virgin olive oil and raw nuts as main fats. Avoids poor quality industrial greases. | 34.5 | 79.3 | 90.8 |
| Has a quality breakfast or lunch, without processed foods | 26.4 | 62.1 | 78.2 |
| Does not peck between meals. | 27.6 | 64.4 | 86.2 |
| Adapts to the food made at home (family) and alternatives not offered. | 37.9 | 64.4 | 85.1 |
| Mealtimes together, avoiding the television or other technology. | 74.7 | 81.6 | 81.6 |
| Regular physical exercise (running, playing, walking, climbing, etc.) or sport. | 69.0 | 70.1 | 81.6 |
| Gets 7–9 h of sleep daily. | 69.0 | 77.0 | 95.4 |

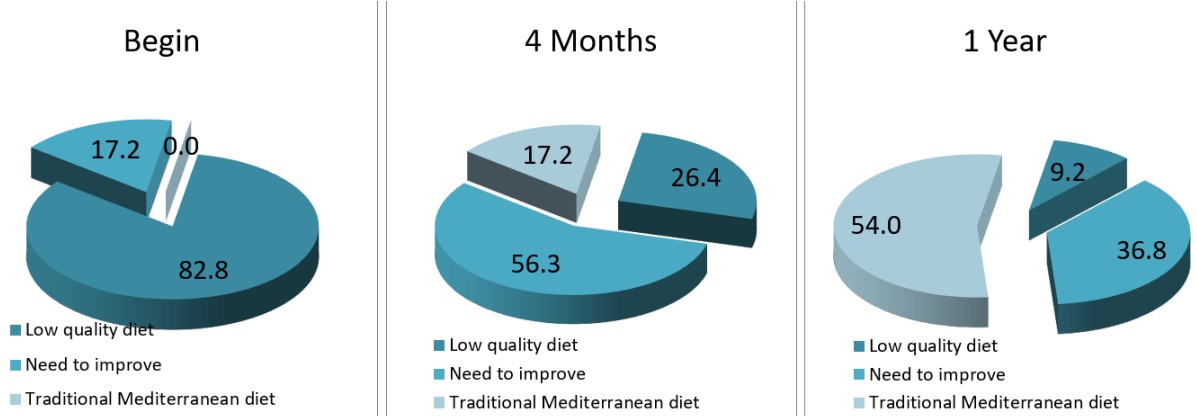

**Figure 1.** Evolution of the quality of diet, measured using the TMD test.

Fourteen of the 15 children had high inflammatory markers measured in the blood, highlighting especially TNF-alpha and, to a lesser extent, IL-1, IL-6, and PCRhs. After the establishment of the TMD, the results were normalized at six months in 13 patients.

## 4. Discussion

In view of the results, we can say that the traditional Mediterranean diet can help prevent and even reduce the exaggerated development of lymphoid tissue suffered by these patients. We have been able to verify that children who have followed our dietary recommendations have improved the inflammatory response and defensive capacity against the usual infectious diseases. During the year of treatment, there was a decrease in the degree of involvement of PNO, which went from a moderate/intense profile to nothing/mild in more than 95.1% of patients. They improved their symptoms of nasal obstruction, including better nasal breathing, decreased nighttime snoring, decreased sleep apnea, and better rest at night. The number of URI with bacterial complications decreased by 80.26% (3.09 from the previous year compared to 0.61 from the year of intervention); 60.9% had no bacterial complications during the year of the study. The use of antibiotics fell by 81.94% (from 2.88 to 0.52 antibiotics/patient/year). Symptomatic treatment decreased by 61.2%. The majority of patients did not require surgical intervention, and clinical evolution suggests that it will no longer be necessary. Emergency visits also decreased, and the degree of family satisfaction was high (Table 2). This better way to defend against prevalent childhood diseases, together with the best anti-inflammatory response, could be the cause of the progressive decline in PNO in our pediatric population. We had difficulties in completing the TMD, since we proposed to carry out a homemade, familiar diet of fresh products that must be elaborated, and the parents did not always have the time and dedication to do it properly. We also limited foods that some parents considered untouchable, such as some foods of animal origin and processed products for breakfast. To alleviate the difficulties involved in carrying out a different type of diet than usual, consultations with

the nutritionist were frequent in order to pave the way towards a quality diet. In our study, the dietary habits of the patients had improved in the whole sample at the end of the program. Thus, an increase was observed in the number of patients who consumed fruits, vegetables, nuts, whole grains, and fermented milk products. In general, the consumption of proteins of animal origin was reduced considerably, especially cow's milk, red meats, and meat products. On the other hand, the percentage of patients who ate industrial bakery breakfast decreased, as well as processed foods.

It is important to note that during the time that the incorporation of patients to the study lasted, we extended the application of the TMD to the entire pediatric population (siblings, relatives, patients with other recurrent pathologies, and infants under two years of age). This led to a progressive decrease in patients diagnosed with PNO, so the achievement of the sample size was delayed. All of this has resulted in the decrease in patients diagnosed with PNO, and it begins to be a rare disease in children of our pediatric quota who follow the TMD. The European study IDEFICS has verified that children who eat an excess of refined flours and processed foods of animal origin, together with infrequent consumption of fruits and vegetables, have high inflammatory markers, which can be considered as a whole and are found in a pro-inflammatory state [29,30]. The results of the inflammatory markers performed in the last years of the study gave very interesting results, since, in the majority of the patients, the figures appeared above normal (pro-inflammatory state), in particular TNF-alpha and, to a lesser degree, IL-1, IL-6, and hsCR. All of these children normalized the inflammatory markers at the end of the nutritional therapy.

By promoting a high-quality diet such as the TMD, we speculate that we are modifying factors that intervene in the proper functioning of the inflammatory and immune system. These could be: (1) a decrease in the antigenic load of the diet, reducing the number of foods with high antigenic power, in particular proteins of animal origin; (2) a balance of the ratio of omega-3/omega-6 polyunsaturated fatty acids, which is strongly deviated in the Western diet in favor of the latter and is a precursor of pro-inflammatory substances; (3) elimination of enzymatic disruptors that are often present in industrial additives; (4) modifications in the composition of the microflora of the oropharyngeal cavity and replenishment of the fermentative intestinal microbiota, as opposed to the putrefactive intestinal flora resulting from the Western diet; and (5) the contribution of anti-oxidants that help control the oxidative stress prevailing in the naso-pharyngeal cavity [31].

Lymphoid tissue associated with mucous membranes (MALT) is composed of cells and lymphoid structures that are organized to provide an adequate defense against foreign agents, microorganisms, toxins, and particles that access the organism through the airway or digestive tract. The most remarkable thing about the immune system is its great ability to discriminate on a molecular scale between what is proper and what is foreign [32]. Secretory IgA constitutes more than 80% of all antibodies produced by MALT, and is responsible for blocking foreign agents [32]. However, its production is very low in the first years of life, reaching normalization no earlier than 10 years [33]. In children older than four years with problems of tubal obstruction, it was observed that they had lower levels of IgA than normal children [34]. It seems that the presence of biofilms on the adenoid surface could be a reservoir of microorganisms that could cause chronic inflammation [35].

The introduction of additional foods is associated with changes in the gut microbiota composition, possibly altering immune and metabolic mechanisms [36]. Recent studies have revealed that consumed foods affect the immune response. The components of these foods act on several immune cell types; their effects are mediated by the intestinal immune system and, in some cases, by the gut microbiota. The improved immune response defends the host against infection and inhibits immune responses, suppressing allergies and inflammation [37,38]. All of this suggests that factors are related to the introduction of infant foods that alter inflammatory and immune mechanisms, favoring the development of PNO.

There are hardly any studies that relate lymphoid hyperplasia with changes in current diet, as repeated infections of the respiratory tract are systematically attributed to it. The

bibliography is focused on the pharmacological and surgical treatment. Currently, ENT specialists tend to administer a cycle of antibiotics for several weeks after diagnosing a PNO, together with oral or local corticosteroids, followed by a three-month observation period. If the symptoms of nasal obstruction persist, with/without effusion in the middle ear and nighttime snoring, they indicate the surgery that best suits each patient [6,39].

The amygdala adenoidectomy, next to the implantation of tympanic drainages, is the most frequently performed surgical technique. Its objective is to achieve an increase in the air space of the nasopharyngeal cavity, together with a normalization of nasal obstruction and its comorbidities [9]. Although this technique has been proven to be effective in resolving the asphyxiating situation caused by the hyperplasia of the lymphoid tissue, it is not free of complications, the most common being bleeding and burns during the cauterization process [8]. In a recent meta-analysis on the effectiveness of adenoidectomy for nasal symptoms, it was concluded that current tests are scattered, inconclusive, and have a significant risk of bias [40]. In another meta-analysis on pediatric sleep apnea, it was concluded that the amygdalo-adenoidectomy does not heal, and that the complete resolution of the apnea-hypopnea index is not achieved, although there was a significant improvement [41]. The exeresis of lymphoid tissue does not seem to have a negative influence on the immune system, although the problems that may arise in the future on the elimination of healthy lymphoid tissue are not exactly known. It is known that they cause postoperative declines of IgG and IgA, with partial recovery in the medium term [42].

One of the characteristics that every research study should have is that it is easily reproducible, uses small groups, and has little economic cost. The work presented here is easy to reproduce in any primary care pediatric consultation, but it is not easy to perform, due to the lack of nutritionists and the lack of effective monitoring of the diet performed. Our study has limitations, in particular the lack of a control group, which would allow us to compare the results. We could not perform a study with a control group, since most of our pediatric space was adhering to the Mediterranean diet, and it did not seem ethical to promote a pro-inflammatory, Western-type diet in a control group.

It would have been very interesting to perform analyses that measured the response of the immune system, inflammatory markers, and data on the modification of the microbiota when making the nutritional change.

Although with age comes a slow spontaneous tendency to resolve PNO, such a rapid disappearance of symptoms is not expected. A notable decrease was found in the number of children who required pharmacological and surgical treatment. We thus deduce that the nutritional intervention was beneficial for them. Before and after studies such as ours are prospective, and provide a moderate level of evidence. Most importantly, the results of this research could support further studies on the influence of the Mediterranean diet on this and other chronic inflammatory diseases.

The present study is only part of a general project that we are carrying out that covers most of the recurrent diseases of childhood. We believe that the diet proposed by Western civilization is the origin of alterations in the inflammatory and immune mechanisms, and therefore the cause of most childhood diseases. Most of our patients have been consecutively included in the program "Learning to eat from the Mediterranean" and, as a result, the prevalence of PNO and other recurrent diseases has decreased considerably [43]. The change of the "model of medicine" that these research studies entail should not go unnoticed. It is no longer about remedying a disease with external drugs outside of the defensive system or about limiting surgical interventions, but the therapeutic proposal is based on providing the body with everything it needs to solve its needs and eliminate that for which it is not ready. We can conclude by saying that the application of the traditional Mediterranean diet could be effective in the prevention and treatment of persistent nasal obstruction, limiting pharmacological and surgical intervention in many of these patients.

**Author Contributions:** Conceptualization, F.M.C.-S. and B.C.; methodology, F.M.C.-S.; software, B.C.; validation, F.M.C.-S., B.C. and A.C.; formal analysis, B.C.; investigation, F.M.C.-S., B.C. and A.C.; resources, F.M.C.-S.; data curation, B.C.; writing—original draft preparation, F.M.C.-S.; writing—review and editing, F.M.C.-S.; visualization, F.M.C.-S. All authors have read and agreed to the published version of the manuscript.

**Funding:** This research received no external funding.

**Institutional Review Board Statement:** The study was approved by the Research Committee of the General University Hospital of Ciudad Real (Internal code: C-95, Act 03/2017).

**Informed Consent Statement:** Informed consent was obtained from all subjects involved in the study.

**Data Availability Statement:** Not applicable.

**Conflicts of Interest:** The authors declare no conflict of interest.

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
