# Peer review of "Persistent Nasal Obstruction: An Expression of the Pro-Inflammatory State?"

_2673-351X, doi:10.3390/sinusitis5010010_

Round 1

Reviewer 1 Report

The topic of the research is highly interesting; a significant number of children was included so that data can be considered reliable. Scientific explanations were given to informations which are generally reported on the importance of following a Mediterranean diet at any age. Tables are highly explicative and present data in a clear way.

The results of the research could support further  studies on the influence of the Mediterranean diet on other chronic inflammatory diseases.

The main question addressed by the authors is that in children persistent nasal obstruction could be prevented and treated by decreasing nasal obstructive inflammation following the traditional Mediterranean diet. The topic is interesting and quite relevant as it highlights the role of children eating habits on the degree of nasal obstruction and the reduction of complications.  The subject is original; usually, the effects of the Mediterranean diet are taken into consideration in obese or type I diabetic children. This is the first study which describes the role of the Mediterranean diet on persitent nasal obstruction. The paper is well written and the study design was well constructed. The writing is fluent and it captures the reader's attention. The conclusions are reported in a straightforward way; the authors underline that data are still preliminary and are part of a general project which they are still undergoing in order to study how to reduce inflammatory diseases in children. Data reported in the study support the authors' suggestion of an existing correlation between decreasing persistent nasal obstruction and decreasing inflammation after following the Mediterranean diet.  I strongly suggest acceptance of the paper.

Author Response

Thank you for your kind comments.

Reviewer 2 Report

Dear Authors,

I found the general concept of the manuscript interesting. The study promotes healthy diet, which is a benefit. I also found the way the manuscript was written sound and clear.

Unfortunately there are aspects that rise serious doubt:

Although there was great improvement in the study group there was no control group to compare. In many cases of children spontaneous resolution of their health problems might appear with time and it difficult to access if in fact only diet had such a major effect. No other factors were taken into account.

No other reasons for the condition of patients was taken into account. For example no allergies. The Mediterranean diet is not a hypoallergic one and there is a different allergy profile in the Mediterranean region, compering to other parts of the world. Another aspect is microbiota of the digestive track, that can also be influenced by diet. I would suggest adding some other positions to the references:

https://pubmed.ncbi.nlm.nih.gov/28657607/

https://pubmed.ncbi.nlm.nih.gov/30410550/

https://pubmed.ncbi.nlm.nih.gov/30886111/

https://pubmed.ncbi.nlm.nih.gov/30044008/

Author Response

Reply to reviewer 2

In red: Reviewer questions

In black: Replies to the reviewer

In purple: Responses to the reviewer that are incorporated into the text

1.- Regarding the control group and the tendency to spontaneous healing with age.

We are a research group in primary care, with few resources and a small number of patients. We have ventured to investigate this area of ​​paediatric nutrition because no one to our knowledge was doing so and it seemed relevant. Our objective is to alert the scientific community about the effects of the Mediterranean diet on inflammatory processes, so that if appropriate, more complete studies such as clinical trials can be performed.

We could not perform a study with a control group since most of our paediatric space was adhering to the Mediterranean diet and it did not seem ethical to promote a pro-inflammatory western-type diet in a control group.(Pag 12)

We have just completed a study, currently in the review phase, that applied a Mediterranean diet to all infants in the sample from birth. We have obtained satisfactory results, and although larger-scale studies are needed, these limited findings open new avenues of research. Inflammatory diseases in general and PNO in particular decreased in the study43.

Although with age comes a slow spontaneous tendency to resolve PNO, such a rapid disappearance of symptoms is not expected. A notable decrease was found in the number of children who required pharmacological and surgical treatment. We thus deduce that the nutritional intervention was beneficial for them. Before-and-after studies such as ours are prospective and provide a moderate level of evidence. Most importantly, the results of this research could support further studies on the influence of the Mediterranean diet on this and other chronic inflammatory diseases. (Pag 12-13)

  1. Allergy

In patients suspected of allergic processes, tests were performed to rule these out. (Pag 3) In our environment, the most frequent allergy is rhino-conjunctivitis against grasses and olive trees. The symptoms are highly characteristic and distinct from persistent nasal obstruction, although when in doubt, we performed studies to rule this out.

Regarding food allergy, the clinical picture also differs, although we think that food and the microbiota could be involved in the pathogenesis of PNO. We have a study on this topic pending publication.

3.- Microbiota

We believe that the Mediterranean diet provides an adequate microbiota to humans living in the Mediterranean area. It seems a very interesting hypothesis that an inappropriate microbiota, generated by the western diet, can cause dysbiosis leading to immune alterations and recurrent inflammatory disorders.

We have added the following text and bibliographic citations:

The introduction of additional foods is associated with changes in the gut microbiota composition, possibly altering immune and metabolic mechanisms1. Recent studies have revealed that foods consumed affect the immune response. The components of these foods act on several immune cell types; their effects are mediated by the intestinal immune system and, in some cases, by the gut microbiota. The improved immune response defends the host against infection and inhibits immune responses, suppressing allergies and inflamation2-3. All of this suggests that factors are related to the introduction of infant foods that alter inflammatory and immune mechanisms, favouring the development of PNO. (Pag 12).

  1. Differding MK, Benjamin-Neelon SE, Hoyo C, Østbye T, Mueller NT. Timing of complementary feeding is associated with gut microbiota diversity and composition and short chain fatty acid concentrations over the first year of life. BMC Microbiol.2020 Mar 11; 20(1):56. doi:10.1186/s12866-020-01723-9. https://bmcmicrobiol.biomedcentral.com/articles/10.1186/s12866-020-01723-9
  2. Uncovering the Immunological Properties of Isolated Lymphoid Follicles. Layhadi JA, Samji MH. Allergy 2020;00:1-2. https://doi.org/10.1111/all.14598
  3. Aitoro R, Paparo L, Amoroso A, Di Costanzo M, Cosenza L, Granata V, Di Scala C, Nocerino R, Trinchese G, Montella M, Ercolini D, Berni Canani R. Gut Microbiota as a Target for Preventive and Therapeutic Intervention against Food Allergy. 2017 Jun 28;9(7):672. doi: 10.3390/nu9070672. PMID: 28657607; PMCID: PMC5537787.

(Pag 15)

Reviewer 3 Report

This is a non blinded, non randomized, non placebo controlled trial of introducing a change in diet and measuring the change in symptoms and need for various treatments of upper airway obstruction in children 2 t o 8 years of age.

The authors state that there was a 95.1% reduction in degree of nasal obstruction. However, this was based on asking them if their nasal obstruction if their symptoms were 0=mild,1=moderate and 2=intense. The mean score improved from 1. 92 to 0.26 although this may have been due to regression to the mean or growth over time. There is no objective measurement of nasal obstruction.

In table 1 most of the improvement in nasal congestion occurred in the first 4 months which suggests the improvement may not have been due to diet. The data in Table 1 should also describe the range and variability of the responses.

The limitations of this type of study are not fully described.

The conclusions that were reached are overstated and the authors should be much more cautious in the interpretation of the results they obtained.

Author Response

Responder al revisor 3

En rojo: preguntas de los revisores

En negro: respuestas al revisor

En violeta: respuestas al revisor que se incorporan al texto

1.- El grado de obstrucción nasal

Se realizó un diagnóstico clínico pediátrico y se evaluó el grado de obstrucción nasal antes, a los 4 meses y al año para evaluar las dificultades en la respiración nasal. Asimismo, se evaluaron síntomas asociados sugestivos de obstrucción nasal persistente (hipertrofia de amígdalas y adenoides, voz nasal, respiración bucal, ronquidos, dificultad para respirar durante el sueño, paro respiratorio al dormir [apnea], sueño inquieto, posturas de hiperflexión del cuello al dormir, somnolencia o de no haber descansado adecuadamente, facies adenoidea y dificultades para tragar). No se realizaron mediciones radiológicas porque no se consideraron relevantes. Priorizamos la sintomatología y resolución de la obstrucción nasal persistente.

We measured persistent nasal obstruction in the three degrees described, and this was evaluated by the paediatrician. We also found it important to assess the nutritional treatment responses of patients and their families, referred to in the clinical response test.

2.- Tendency to regression to the mean with age

Although with age comes a slow spontaneous tendency to resolve PNO, such a rapid disappearance of symptoms is not expected. A notable decrease was found in the number of children who required pharmacological and surgical treatment. We thus deduce that the nutritional intervention was beneficial for them. Before-and-after studies such as ours are prospective and provide a moderate level of evidence. Most importantly, the results of this research could support further studies on the influence of the Mediterranean diet on this and other chronic inflammatory diseases.(Pag 12-13)

3.- Most of the improvement in nasal congestion occurred in the first 4 months, suggesting that the improvement may not have been due to diet. The data in Table 1 should also describe the range and variability of the responses.

A month after starting the Mediterranean diet, we began to see a better defensive disposition against recurrent catarrhal diseases, reducing their presentation and bacterial complications, which facilitated a decrease in inflammation and better nasal breathing. It was not possible to modify the data in table 1.

  1. The limitations of this type of study are not fully described.

We are a research group in primary care, with few resources and a small number of patients. We have ventured to investigate this area of ​​paediatric nutrition because no one to our knowledge was doing so and it seemed relevant. Our objective is to alert the scientific community about the effects of the Mediterranean diet on inflammatory processes, so that if appropriate, more complete studies such as clinical trials can be performed.

We could not perform a study with a control group since most of our paediatric space was adhering to the Mediterranean diet and it did not seem ethical to promote a pro-inflammatory western-type diet in a control group. (Pag 12)

We have just completed a study, currently in the review phase, that applied a Mediterranean diet to all infants in the sample from birth. We have obtained satisfactory results, and although larger-scale studies are needed, these limited findings open new avenues of research. Inflammatory diseases in general and PNO in particular decreased in the study.

5.- The conclusions that were reached are overstated and the authors should be much more cautious in the interpretation of the results they obtained.

Podemos concluir diciendo que la aplicación de la Dieta Mediterránea Tradicional es eficaz en la prevención y tratamiento de la obstrucción nasal persistente con una disminución significativa de la inflamación obstructiva nasal, lo que ha determinado que la mayoría de los pacientes han mejorado clínicamente y no precisarán intervención quirúrgica.

Podemos concluir diciendo que la aplicación de la Dieta Tradicional Mediterránea podría ser eficaz en la prevención y tratamiento de la obstrucción nasal persistente, limitando la intervención farmacológica y quirúrgica en muchos de estos pacientes. (Pag 1,13)

Round 2

Reviewer 2 Report

Thank you for your response to the revision. I feel that the manuscript improved.